# SMetaS: A Sample Metadata Standardizer for Metabolomics

**DOI:** 10.3390/metabo13080941

**Published:** 2023-08-12

**Authors:** Parker Ladd Bremer, Oliver Fiehn

**Affiliations:** 1Department of Chemistry, University of California, Davis, CA 95616, USA; plbremer@ucdavis.edu; 2West Coast Metabolomics Center for Compound Identification, UC Davis Genome Center, University of California, Davis, CA 95616, USA

**Keywords:** metadata, meta-analysis, repository, FAIR, standardization

## Abstract

Metabolomics has advanced to an extent where it is desired to standardize and compare data across individual studies. While past work in standardization has focused on data acquisition, data processing, and data storage aspects, metabolomics databases are useless without ontology-based descriptions of biological samples and study designs. We introduce here a user-centric tool to automatically standardize sample metadata. Using such a tool in frontends for metabolomic databases will dramatically increase the FAIRness (Findability, Accessibility, Interoperability, and Reusability) of data, specifically for data reuse and for finding datasets that share comparable sets of metadata, e.g., study meta-analyses, cross-species analyses or large scale metabolomic atlases. SMetaS (Sample Metadata Standardizer) combines a classic database with an API and frontend and is provided in a containerized environment. The tool has two user-centric components. In the first component, the user designs a sample metadata matrix and fills the cells using natural language terminology. In the second component, the tool transforms the completed matrix by replacing freetext terms with terms from fixed vocabularies. This transformation process is designed to maximize simplicity and is guided by, among other strategies, synonym matching and typographical fixing in an n-grams/nearest neighbors model approach. The tool enables downstream analysis of submitted studies and samples via string equality for FAIR retrospective use.

## 1. Introduction

### 1.1. Motivation

There is growing interest in standardizing metabolomics data [1,2,3,4]. Such standardization could lead to dramatic increases in biological and biomedical applicability of metabolomics. For example, decreasing the workload involved in the meta-analysis of metabolomics datasets would validate metabolomics’ reproducibility overall, in addition to providing conclusions for those systems which are specifically studied [5,6]. Another way is the development of a vast metabolomics dataset, which would serve as the input to large machine learning models [7]. These multivariate models could aid or even supplant hypothesis-driven biology in the same way that statistical language models reproduce language despite the absence of a comprehensive language theory [8].

There are multiple areas where metabolomics standardization is being improved [9,10]. Areas that we do not address in this work include observational/chemical data acquisition and data processing. In this area, there has been much progress, and it is hoped that efforts toward (semi) quantitation and homogenization of methods will occur. 

We focus here on the development of tools to standardize the metadata that describe samples. Our goal is to enable meta-analysis that occurs on a sample-level and is programmatic. This *programmatic meta-analysis* means the ability of computers and their users to aggregate samples very quickly and very easily. We envision users to be able to aggregate samples by checking the equality of strings (e.g., “species” = “*mus musculus*”) rather than aggregation using natural language tasks. Likewise, we employ ontological relationships (e.g., “X is a type of Y” relationships) to group sample metadata to query the database on different levels of abstraction. 

By emphasizing the sample-level for meta-analysis, we dramatically increase the number of ways that samples can be compared. In traditional meta-analysis, researchers are constrained to explore only the original intentions of authors’, i.e., based on study design factors and hypotheses. If, instead, samples are labeled by every column header and corresponding values (such as body mass index, sex, age), then researchers could reuse those samples to explore any number of new hypotheses therein as a potential on-the-fly-factor (e.g., comparing metabolomes of specific organs across age groups, or organs across diseases). 

### 1.2. Sample Metadata in -Omics

Ultimately, all -omics analyses are based on samples. Challenges to capturing sample metadata from other fields may therefore inform solutions for metabolomics. There is a growing interest in the reuse of sample data for understanding reproducibility of findings, validation of hypotheses or as input into machine learning models. Many projects, databases, or consortia operate by formalizing and mandating metadata standards [11,12,13,14]. While the intent for project-wise metadata standardization is an appealing first step, the dispersion of authority creates challenges for system and data interoperability. It is very difficult to merge databases with different standards in a traceable and logical manner. Today, there are over 1000 metadata standards lodged at https://fairsharing.org [15]. Formalizing and harmonizing these standards is an area of active research, and sophisticated informatics schemes have been proposed to reduce this bottleneck [15,16]. 

Perhaps an even greater challenge is the latency of biological and biomedical communities to adopt metadata standards and to adhere to reporting guidelines. There are at least three obstacles: (a) Definitions of ‘minimum requirements’ and ‘best practice’ surely change over time and between sub-communities. (b) Individual biologists or biomedical researchers do not have immediate benefits or incentives to adhere to metadata standards. This problem may be viewed as a variant of the ‘tragedy of commons’ [17]. (c) Many metadata upload tools are written with underlying database architectures in mind, not with user friendliness. In industry, user friendliness for web interfaces is a primary objective. In academia, the user friendliness of frontends is claimed, but not tested or proven. 

Hence, classic databases and sample submission interfaces expect users to submit samples and their metadata in good faith. The ideas of reusing data have been commonplace since the early 2000s, however, with little progress so far [18]. Attempts to reuse genomics data for COVID-19 analysis revealed that, despite relatively simple requirements, over 77% of 12,000 COVID sequencing experiments lacked location metadata [17]. Similar findings have been reported for metabolomics sample metadata [19]. Finally, as we have recently learned from our own BinBase metabolomics database [20], retroactively assigning machine-ready metadata is either very tedious or demands much more research [20,21].

SMetaS development was therefore focused on the user-facing aspects of sample metadata. We avoided developing yet another standard, and rather created a tool that others can adapt and utilize within their own pipelines. Second, SMetaS intends to simplify the process of creating machine-ready metadata for the non-coding scientist. We take out the user awareness of standards and ontologies and instead employ these as backend for programmatic curation of user-based metadata. We focus on presenting a familiar tabular format, which might increase the fraction of scientists who are willing and able to describe their samples in some detail. 

### 1.3. Sample Metadata in Metabolomics—Tool Critiques

Ultimately, successful programmatic meta-analysis, especially on a community-wide level, becomes an engineering problem. Design choices can improve or hinder a tool’s capacity to facilitate programmatic meta-analysis. We explore and critique three community tools that focus on sample metadata.

### 1.4. Tool Critique—Metabolomics Workbench

On the Metabolomics Workbench platform, samples are submitted as part of a study [22]. A user chooses a single type of sample (human, plant, material, etc.), and is then exposed to a set of sample metadata categories with freetext fields that depend on the chosen sample type (e.g., plant will yield “watering” options, human will not). Additionally, users are exposed to a copy/paste tsv parser for accompanying tabular sample metadata. The Metabolomics Workbench offers design choices that support programmatic meta-analysis. The displayed fields for steps are based on the selections of previous steps, which reduces visual complexity for the user. 

Yet, Metabolomics Workbench also makes design choices that are not favorable for programmatic meta-analysis. The most unfavorable is the use of freetext for sample metadata values (e.g., in the specific headers and in the additional matrix). Until natural language models become much more advanced, this design choice prohibits inter-study programmatic meta-analysis. Similarly, there are too many metadata categories offered to users. Exposure to too many fields easily overwhelms users and reduces metadata accuracy. Finally, Metabolomics Workbench assigns all sample-specific values to the underlying overall study. If a study is annotated as a “human study”, it precludes accurate metadata assignments of other samples within the study, e.g., an integrated comparison of metabolomic profiles of human plasma, food and intestinal bacteria. Instead, SMetaS assigns metadata to each sample separately. We describe this metadata assignment as *sample-level granularity*.

### 1.5. Tool Critique—ReDU

In ReDU, sample metadata are submitted retrospectively to be associated with mass spectra that are uploaded to the GNPS/MassIVE environment [23]. Users copy a Google Sheets template which offers a fixed set of sample metadata categories. For each sample, for each category, users select a value from a finite set of options provided in a dropdown or on another sheet. Completed metadata files can be checked with a graphical tool and then uploaded.

ReDU offers design choices that support usage and programmatic meta-analysis. First, constrained vocabularies enable programmatic meta-analysis via string equality rather than entity recognition. Second, by adopting the concept of sample-level granularity, ReDU expands its capabilities for meta-analysis. For meta-analyses, samples can be selected across studies instead of relying on a (much smaller) subset of studies that shared an overall identical set of metadata. Third, Google sheets are a familiar tool to users, minimizing complexity and encouraging its use. Fourth, there is an ongoing update capability because the metadata categories/headers can be expanded via Github requests.

Unfortunately, ReDU makes design choices that hinder programmatic meta-analysis. Importantly, entries are restricted to specific metadata categories and their corresponding vocabularies, including unclear or rarely used metadata headers like ‘Altitude’ or ‘TermsOfPosition’. It remains unclear to users how such metadata might map to their specific studies. An overlap in the meaning of metadata categories, such as comorbidity and disease, requires post hoc metadata curation before final analyses. This complexity hinders programmatic meta-analysis because future programmers will have to explore and compare vocabulary spaces. Finally, the constrained vocabulary terms are not expandable in an easy way, which precludes sample submission.

### 1.6. Tool Critique—MetaboLights

In MetaboLights, samples are submitted as part of a study [24]. Users are walked through a series of steps that include submitting study-wide attributes and experimental methods. Ultimately, users are exposed to a step where samples are described via an uploadable or buildable sample metadata matrix. 

MetaboLights includes design aspects that favor programmatic meta-analysis. The step-by-step walkthrough simplifies the submission process which encourages use. Likewise, the emphasis on sample-level resolution is essential to FAIR programmatic meta-analysis. Additionally, the connection of metadata categories/headers to ontology terms generates a constrained vocabulary that enables string equality comparisons and that supports downstream ontological analysis. 

Unfortunately, MetaboLights metadata uploads face several problems that hinder programmatic meta-analysis. All ontologies are accessible at any place in the sample matrix. Consequently, tedious downstream metadata merges are required to remove inconsistencies because the same metadata concept can appear in different forms in different ontologies. Likewise, terms can be entered as freetext, which increases the problem of post hoc metadata curation. Finally, uploading all final metadata to the MetaboLights interface requires using an ftp service, which greatly discourages use.

### 1.7. SMetaS

We therefore aimed to create a tool that enables sample-oriented and programmatic meta-analysis if used in a frontend for study submissions to metabolomic databases. Such tools necessarily must remain a compromise between asking users to detail every aspect of a study (e.g., the exact composition of chow in studies of animal models) versus the time and effort users are willing to spend on sample or data submissions. Such tools offer programmatic relationships to existing standards, but do not complexify the space of already existing standards. Indeed, we designed a tool that captures the essence, but not the total complexity, of a sample’s nature, while giving users the option to add more details if they are inclined to do so. This mixture of mandatory and voluntary metadata is auto-curated and recorded into a database, which can then be linked to the observed metabolomics of a sample in downstream analyses.

## 2. Materials and Methods

An overview of the workflow is shown in Figure 1. A more detailed analog is available in Appendix A. For generating the vocabularies and associated models, we made extensive use of custom python scripts that are available in Github (see Data Availability). We heavily employed snakemake, networkx, pandas, scikit-learn, and other libraries [25,26,27]. The API and frontend were also generated using custom python scripts that are available in Github (see Data Availability). We heavily employed Flask, Dash, and Docker. Development and creation were performed locally on a personal computer.

## 3. Results

### 3.1. Overview

The primary result of this work is a tool that facilitates the standardization of sample metadata for downstream programmatic analysis. Basic usage for SMetaS is illustrated in Figure 2. Here, the user first chooses metadata that are associated with their samples. There is no capability to specify “factors” because that is an artificial constraint that can be readily applied downstream. User selections generate a downloadable csv file for which each row is a sample, and each column is a metadata attribute. We chose this format because all scientist users are familiar with such basic worksheets and know how to manipulate these documents. Cells can remain empty if that attribute does not apply. Users then reupload their csv files, and interact with the SMetaS transformation process, which converts freetext, natural language entries into a formalized and standardized representation.

This tool’s most noteworthy design choices that facilitate this are shown in Table 1. The tool is provided as a container that is directly runnable. Associated code is available for the vocabulary pipeline as well as downstream API/frontend. Documentation is available as well.

### 3.2. Extended Descriptions of Components

Our tool is comprised of two main components. In the first component, users walk through a short series of steps where they generate a csv spreadsheet onto which they transcribe their sample metadata. Users can select core sample types (tissue, cells, etc.) as well as additional sample attributes. Users then fill out their created spreadsheet locally and resubmit it to the second part of the tool.

In the second component, the user submission is transformed into an equivalent sample metadata matrix that is ready for programmatic meta-analysis at a later date. To do this, there are several steps that are taken at the user’s submission. By the end of these passes, it is guaranteed that all terms will be transformed into an existing term or become new terms in the corresponding vocabulary. 

In the first pass, a term-frequency-inverse-document-frequency (tf-idf) vectorizer and nearest neighbors model (nnm) automatically curates user-submitted strings [28,29]. This vectorizer works by transforming a given string into a numeric vector and then finding the vocabulary term with the most similar vector. The components of the numeric vector are decided during the database construction step in Figure 1/Appendix A. The vector has components of all three-character combinations present at least once. For example, *mus musculus* would generate (m,u,s), (u,s,’ ‘), (s,’ ‘,m), (‘ ‘,m,u), (u,s,c), etc., while *arabidopsis thaliana* would generate (a,r,a), (r,a,b), etc., and the union of each term’s set generates the total set of vector components. The magnitudes of each cell in the term/component matrix are based on presence of that component in a term after weighting component magnitudes according to the number of appearances within that term (term frequency) and rareness of that letter triplet across all terms (inverse document frequency). New words are coerced into this pre-determined space and cosine similarity determines the distance between vocabulary terms and user-provided terms.

The tf-idf vectorizer and nnm curate strings such as species (e.g., mice, mouse, M. musculus and similar terms), organ names, drug names, units, or other metadata. Metadata that are intrinsically unique to a sample (e.g., magnitudes of height or drug amount) are not curated. The derivations of the initial controlled vocabularies from official ontologies are described in Table 2. The first pass is expected to deal with the bulk of user-submitted strings. 

In the second pass, terms that were not able to be mapped in the first pass can be transformed using a substring search. This might happen if the term was accidentally misspelled, or if an unknown synonym or abbreviation was used. Both the first and second passes map sets of strings to “main terms”. For example, “*mus musculus*”, “mouse”, “mice”, or “house mouse” would all be mapped to “*mus musculus*”.

Finally, in the third pass, users can confirm to add new terms that were not present in the associated header’s vocabulary, for example, for organ, species or experimental intervention that were not included in the large, standardized community vocabularies that we employ (see below). In this way, users add new strings to our underlying ontologies to update and renew the system over time to increase the likelihood that the next users will find matching selections (e.g., for new cell types, drugs, etc.). Users are given a freetext input box preloaded with the observed term. Once confirmed, these freetext terms are added to the corresponding vocabulary for future users, and corresponding models will be retrained on this expanded vocabulary.

Users receive a standardized csv file to be used to submit a study for metabolomic data acquisitions (e.g., at the UC Davis West Coast Metabolomics Center), or to submit metabolomic data to a common repository (e.g., for the MetabolomicsWorkbench) to enable programmatic meta-analysis. The system includes a programmatic access point for submitted studies and authors for convenient integration into existing pipelines.

### 3.3. Use Case

We provide here an explicit example based on a study performed at the West Coast Metabolomics Center involving the effect of ozone on metabolism in the lung [30]. In Figure 3, we show an excerpt from the study abstract, the freetext representation created with our tool, and finally, the curated representation created with our tool.

Briefly, male and female adult mice were exposed to house dust mite allergen, then exposed to ozone, and lastly, sacrificed; differences in lung metabolism were compared to untreated control mice by metabolomics assays. Most of the information given in the publication (Figure 3a) was captured by SMetaS, but not all details of the study design (Figure 3b). We recorded basic descriptions of the lungs (organ, mass, massUnit), the mice from which they were derived (species, sex, age, ageUnit, strain), the treatment of ozone exposure represented as a drug (drugName, drugDoseMagnitude, and drugDoseUnit), and the time-series aspect after exposure to the allergen (zeroTimeEvent, time, and timeUnit). We lose information such as the intranasal delivery, ozone chamber details, and high-detail lung lobe locations. We recognize that there are other valid ways to represent this study. For example, it would have been possible to represent the allergen exposure as another drug. Indeed, creating unambiguous instructions for metadata representation is an area of active research [15].

Importantly, SMetaS transforms freetext strings to formalized nomenclature (Figure 3c) by mapping to pre-existing terms. All non-numeric strings were already contained in the initial ontology-derived vocabularies except for ‘allergen exposure’, ‘ozone’, and ‘hours/day’ and freetext strings were successfully mapped. These three metadata strings were then added to their corresponding vocabularies (zeroTimeEvent, drugName, drugDoseUnit) for future users.

### 3.4. Construction of Vocabularies

Sample metadata standardizers should incorporate vocabularies. As expanded on in the discussion, there are several important properties of vocabularies. SMetaS relies on a constrained, non-overlapping, non-redundant, and expandable vocabulary for each metadata category. Constrained vocabularies allow for equality testing via string equality rather than entity recognition. For example, for databases supported by SMetaS, all samples associated with mice are mapped to “*mus musculus*”. For databases that accept freetext without automatic curation, users must devise post hoc models or elaborate search criteria to collect those mice samples. Such post hoc data curation easily creates errors, dramatically increases the workload, and decreases overall data quality.

We limited the number of downloaded vocabularies and ontologies to large, mature community repositories such as MeSH [31], NCBI [32,33], Cellosaurus [34], NCIT [35], and FDA [36]. These non-redundant vocabularies avoid string overlaps and therefore abolish the need for complicated downstream merges between headers and terms. Finally, expandable vocabularies minimize system maintenance and allow for perpetual updates when users submit, for example, a new drug, genetic variant, animal model, etc. 

To generate vocabularies that maintain our first three principles, we accessed a set of ontologies and vocabularies relevant to metabolomics study samples, listed in Appendix A. For each header/category, we extracted sets of non-overlapping “main” vocabulary terms, each of which had a set of 0 to n synonyms derived from the same sources. For example, “*mus musculus*” would be a main term, while “mouse”, “mice”, or “house mouse” would be synonyms. The selections for vocabulary origins were made based on internal discussion. The headers and vocabularies are summarized in Table 2.

## 4. Discussion

### Shortcomings and Future Developments of SMetaS

There are several shortcomings of the current SMetaS design. Intrinsic problems are derived from the decision to use a tabular approach to retaining study metadata information. No matter which set of metadata categories we offer to users, eventually there may be a study that cannot be fully described using that schema. The study design space is extraordinarily broad, and we have encountered surprising factors such as “proximity to parking lots” in studies submitted to our metabolomics service core at UC Davis. No tractable tabular approach with a fixed set of metadata could capture such design. Instead, we focus on capturing the essence of a sample rather than its full intricacies. In doing so, we assume that metadata categories that we do not include do not define different populations in a statistical sense. 

We expect to improve the sample description space of SMetaS based on samples submitted to the West Coast Metabolomics Center because our service core analyzes over 30,000 samples/year. SMetaS offers the metadata categories ‘comment’, ‘inclusion criteria’, and ‘exclusion criteria’ as alternatives to categories that are explicitly provided. Over time, we will accumulate data on metadata objects that are common enough to warrant creating explicit categories. Such commentary metadata also give us insights which exiting categories or units *do* cause discrepancies in study descriptors that are insufficiently covered yet. For example, in metagenomics, nuanced aspects of environment or host descriptors are necessary for valid interpretations [17]. Hence, future versions of SMetaS will include additional categories/vocabularies derived from submitted data. 

The tabular approach in SMetaS is also limiting the relationship between descriptors and samples. For example, we assume unique “is a” relationships between a sample and its listed species. While our approach can handle multiple species (for the user, simply a delimiter is needed), the precise meaning of a list of species is not specified. To add additional relationship types (such as co-cultures) would mean increasing the complexity of the written string, which is philosophically prohibited in our fixed vocabulary approach (of course, we cannot prevent users from submitting messy custom terms). Future improvements would maintain sample metadata in node/edge graphs, where the edges afford the opportunity to programmatically store more detailed relationships between samples and descriptors. We illustrate this in the transition from Figure 4a to Figure 4b. 

Storing sample metadata as graphs would simplify querying data in hierarchical and ontology-based meta-analysis. As illustrated in Figure 4c, we see that ontologies map neatly with sample sets when using a graph approach and we can imagine this implementation affording very natural queries. 

While tempting, the notion of a parser that automatically derives sample metadata from natural language is likely not applicable to existing publications or existing datasets for which the assignment of unique properties to specific samples is not clear. 

In general, we recognize some of the shortcomings that are built into our system. We accept these as an inevitable consequence of our goal to render this tool intuitive for users. Challenges and complexities have been documented when users deposit studies into repositories [19]. While our tool aims to conform to user expectations with a frontend that is intuitive enough to be completed without further manuals or communications, our tool has not yet been deployed as a mandatory study submission frontend for a metabolomic repository. User feedback is currently sought from users of the UC Davis metabolomics service core with its approximately 300 clients/year. Upon completion, an updated frontend will feed into the new UC Davis LC–BinBase system, replacing the outdated miniX (SetupX) system that was in operation for 18 years [37].

## 5. Conclusions

We have created SMetaS, which standardizes submitted sample descriptions to enable downstream programmatic meta-analysis. This tool is readily deployable in a fully reproducible way for core laboratories or larger repositories.

We are interested in standardizing metadata to programmatically utilize metadata. We envision at least two ways of doing this en masse. The first is agglomerative analysis similar to those available in our BinDiscover tool [20]. There, we allowed for the exploration of combinations of metadata to be visualized and explored according to user-specified nodes on ontological hierarchies. We can imagine expanding upon this concept and programmatically probing which data patterns persist when comparing studies and samples on their highest ontological parent nodes, i.e., what are the largest generalizations that can be made? Validations of such meta-analyses would be based on published ground truths (e.g., absence of cholesterol in the plant kingdom), giving credibility to new hypotheses to be revealed by large scale database queries. 

The second way that we hope to utilize this tool is as the foundation for a comprehensive metabolomics atlas. We hope that this atlas could be one of many large, normalized datasets provided to a large machine learning model that circumvents hypotheses altogether to provide clinical or therapeutic predictions in opposition to or in conjunction with theory provided by domain experts.

## Figures and Tables

**Figure 1 metabolites-13-00941-f001:**
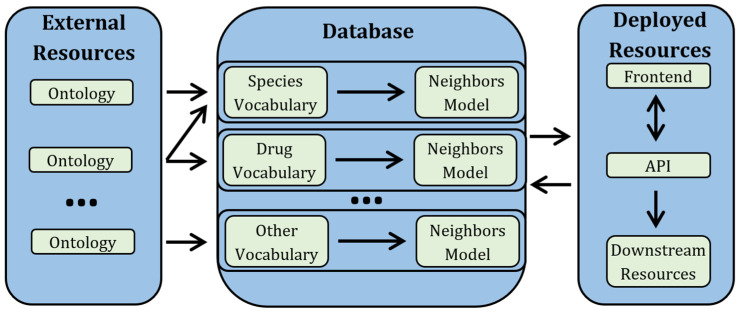
Workflow of SMetaS. First, each metadata column type (species, organ, drug, etc.) has a starting vocabulary derived by combining/subsetting existing ontologies/vocabularies, making sure that the intersection of any two vocabularies is 0. Next, for each vocabulary, we generate additional resources that facilitate ease-of-use for sample submitters (e.g., nearest neighbor models that map synonyms/typos to the correct term). Finally, we make the vocabularies and associated resources as the backend to a user-friendly frontend. These vocabularies and models are expandable if new terms are desired by users. The vocabularies and models are also available as an API directly. A more detailed workflow is available as Appendix A.

**Figure 2 metabolites-13-00941-f002:**
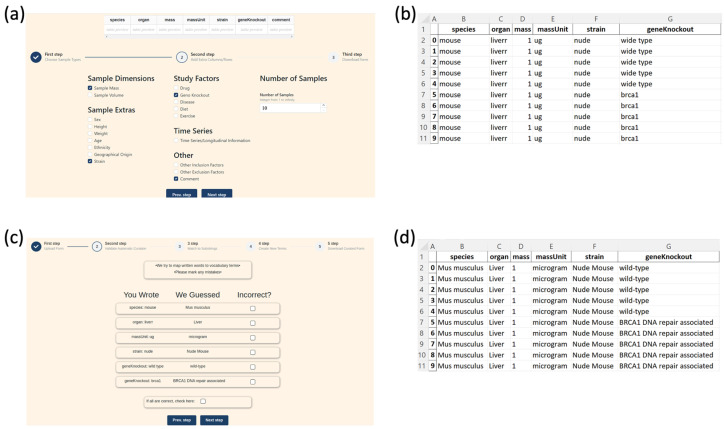
Walkthrough of user experience. (**a**) The first component of the tool is a walkthrough that allows users to design a sample metadata matrix. (**b**) An example metadata matrix prior to standardization. (**c**) The second component of the tool is a walkthrough that allows users to curate that submission. (**d**) The same submission with terms standardized to simplify meta-analysis.

**Figure 3 metabolites-13-00941-f003:**
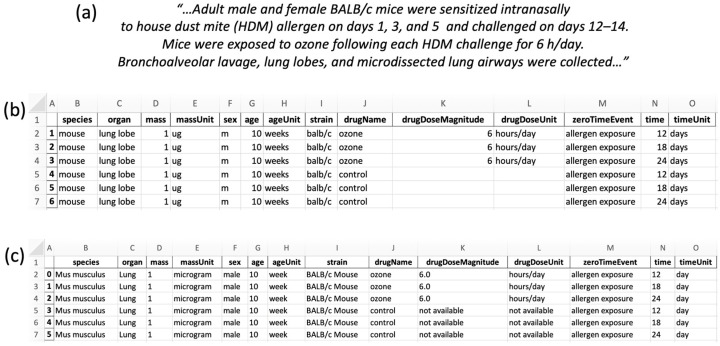
Example use case of SmetaS. (**a**) Excerpt of a published study abstract [30]. (**b**) SMetaS matrix representation of information from the abstract and methods section [30]. (**c**) SMetaS curation of freetext terms of the matrix representation.

**Figure 4 metabolites-13-00941-f004:**
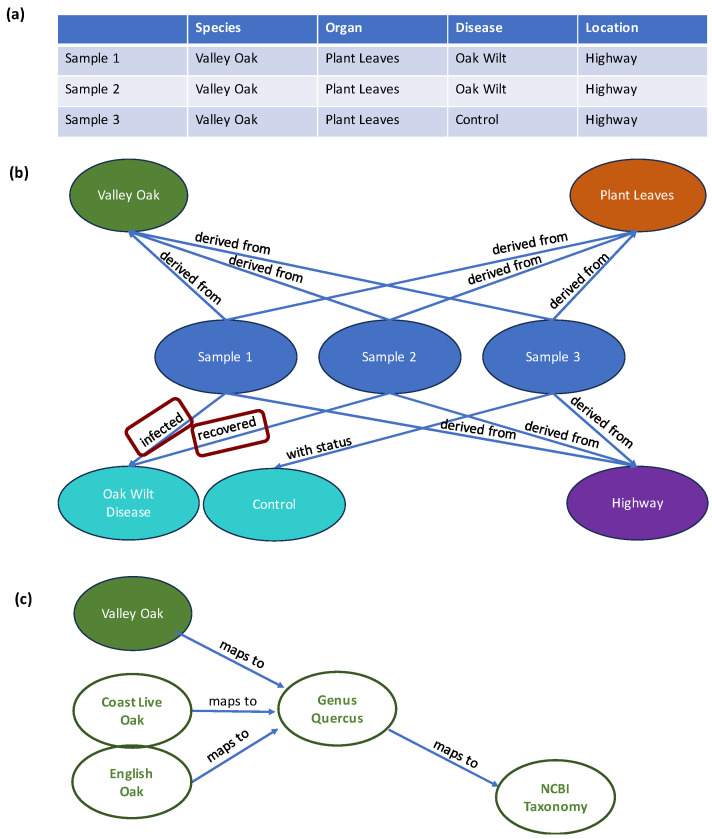
Comparing tabular to graph storage for sample metadata. (**a**) Example of a simple tabular schema for metadata capture. (**b**) The same tabular schema expressed as a graph, with additional complexity programmatically embedded into nodes. (**c**) Example how graph nodes, *here:* the species node ‘Valley Oak’, map to ontologies that can later be used for hierarchical analyses of metadata.

**Table 1 metabolites-13-00941-t001:** SMetaS Design Principles.

Number	Design Principle
1	headers with orthogonal vocabularies
2	vocabularies with non-redundant terms
3	inclusion of a synonym set for each “main term” to facilitate the loose expression of a term
4	vocabularies/models that expand to incorporate new terms easily submitted by users
5	machine learning models that increase speed-of-use and make the program typo tolerant
6	a deference for simplicity when possible, because we believe that user apathy/disinterest is as much a problem as any technical challenge

**Table 2 metabolites-13-00941-t002:** The metadata categories, term counts, and definition of initial vocabularies. The Initial Vocabulary Description column describes what subsections of formal ontologies comprise each vocabulary, initially.

Grouping	Metadata Category	Term Count	Initial Vocabulary Description
Core Sample Type	species	724,962	NCBI ontology less -rank ‘strain’ -parent node scientific name contained ‘environmental sample’ -parent node scientific name contained ‘unclassified’ -rank ‘no rank’ that contained ‘/’ -rank ‘species’ containing numerical characters -rank ‘species’ containing ‘vector’
	organ	11,494	MeSH ontology heading ‘A’ and lower
	cellLine	247,365	Cellosaurus ontology
	material	2056	MeSH ontology: -heading ‘D20’ and lower -heading ‘G16’ and lower
Sample Description	massUnit	49	Unit Ontology: -heading UO0000002 and lower
	volumeUnit	79	Unit Ontology: -heading UO0000095 and lower
	sex	3	All sexes
	heightUnit	48	Unit Ontology: -heading UO0000001 and lower
	weightUnit	49	Unit Ontology: -heading UO0000002 and lower
	ageUnit	22	Unit Ontology: -heading UO0000003 and lower
	ethnicity	1057	NCIT Ontology: -header C17049 and lower
	geographicalOrigin	799	MeSH ontology: -header Z01 and lower -header G16.500.275 and lower
	strain	2282	NCIT Ontology: -header C14250 and lower except those terms which exist in the NCBI ontology or are descendants of Gene header in NCIT ontology
Study Factors	drugName	9537	FDA drug vocabulary
	drugDoseUnit	753	Unit Ontology
	geneKnockout	141,605	NCBI human gene vocabulary
	disease	36,378	MeSH ontology: -header C and lower
	diet	1164	MeSH heading G07.203 and lower
	exercise	569	MeSH heading I03 and lower, MeSH heading G11.427.410.698 and lower
Time Series	zeroTimeEvent	69,321	NCIT: ontology: -header C43431 and lower
	timeUnit	22	All Units

Other	inclusion	0	None in initial vocabulary
	exclusion	0	None in initial vocabulary
	comment	0	None in initial vocabulary

## Data Availability

All code is available at https://github.com/metabolomics-us/metadatastandardizer, accessed 9 August 2023. We make extended documentation available at https://metabolomics-us.github.io/metadatastandardizer/. The documentation reviews deployment instructions for this tool on Amazon Web Services as well as a detailed walkthrough for using the backend.

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
