# Peer review of "SMetaS: A Sample Metadata Standardizer for Metabolomics"

_metabolites, 2023, doi:10.3390/metabo13080941_

Round 1
Reviewer 1 Report
In the present paper the authors reported a user-centric tool (SMetaS) that automatically standardize sample metadata. The tool will be particularly suitable for data reuse, meta-analyses, cross-species analyses and to build large scale metabolomic atlases. User can designs a sample-metadata matrix that the tool then transform and simplify to keep the data consistent and useful.
Minor:
At pag 9 Line 290 a reference is missing.
The tool developed by the authors could be a great improvement in metabolomic field.
Looking forward to adding this in our lab pipelines.
Author Response
critique: At pag 9 Line 290 a reference is missing.
Reply by authors: Fixed
Reviewer 2 Report
Authors present a tool to standardize meta data and its collection for the metabolomic data samples.
It is a useful contribution, knowing that biological data samples may have many features inclusion of which in an orderly way into a sample metadata may enrich future data analysis and comparisons.
However, this article needs more work. Introduction must review existing approaches and lead to a better explained rationale for the presented tool. Currently existing approaches are presented in the end as discussion. Second, you use specific natural language processing terminology that needs at least a short explanation. For example Page 4 Line 123 “ term-frequency-inverse-document-frequency vectorizer and nearest neighbors model automatically curates...” . Please introduce terminology to the reader of the article and provide references. It is not clear how the “term-frequency-inverse-document-frequency vectorizer” curates species and organ names. And third, this work would benefit greatly if the Methods section was included with a concrete actual use case example.
Other comments:
1. Tools that are described in discussion in my humble opinion must be introduced and described in the Introduction leading to the rationale of why your tool was necessary to develop.
2. You frequently mention a programmatic meta-analysis. It would help if it would be introduced in the introduction. Also page 7 lines 203-207 are packed with the specific terminology that again IMHO must be introduced , for example :
“programmatic meta-analysis via string equality rather than entity recognition. Second, the sample-level granularity naturally expands meta-analysis capabilities by enabling a looser selection of samples across studies, rather than forcing meta-analysis to focus on the small subset of comparisons that can be made on those studies with the same hypothesis” .
Here it would be helpful if you could give an example of ‘string equality’ versus ‘entity recognition’. Or explained it better. Or ‘sample level granularity ‘ - what does that mean?
3. Page 5 Table 2 – the Initial Vocabulary Description column. Please explain what it represents, not clear from the text.
4. Page 9 Line 290 – incomplete reference.
Author Response
Critique “Introduction must review existing approaches and lead to a better explained rationale for the presented tool. Currently existing approaches are presented in the end as discussion.”
- Tools that are described in discussion in my humble opinion must be introduced and described in the Introduction leading to the rationale of why your tool was necessary to develop.
Reply by authors: We have rearranged the manuscript such that other tools are now in the introduction. We have also extended both introduction and discussion, with proper references.
Critique: specific natural language processing terminology that needs at least a short explanation. For example Page 4 Line 123 “ term-frequency-inverse-document-frequency vectorizer and nearest neighbors model automatically curates...” . Please introduce terminology to the reader of the article and provide references. It is not clear how the “term-frequency-inverse-document-frequency vectorizer” curates species and organ names.
Reply by authors: We have expanded the text surrounding this concept, including proper references.
Critique: this work would benefit greatly if the Methods section was included with a concrete actual use case example.
Reply by authors: We have added a concrete use-case to the discussion section. Here, we used a published study (from our own lab) that had two different exposures on a mouse model, 2 sexes and a longitudinal dimension of exposures. We clarify the benefits and limits for SMetaS using this case.
Critique: You frequently mention a programmatic meta-analysis. It would help if it would be introduced in the introduction.
Reply by authors: We have added explanation of that term to the introduction.
Critique: Here it would be helpful if you could give an example of ‘string equality’ versus ‘entity recognition’.
Reply by authors: We have expanded the text surrounding this concept.
Critique: ‘sample level granularity ‘ - what does that mean?
Reply by authors: We have expanded the text surrounding this concept.
Critique: Page 5 Table 2 – the Initial Vocabulary Description column. Please explain what it represents, not clear from the text.
Reply by authors: We have added text in the table caption.
Critique: Page 9 Line 290 – incomplete reference.
Reply by authors: Fixed
Reviewer 3 Report
- In this manuscript, the authors describe SMetaS, a tool to standardize sample metadata. Overall, I think the potential of this program to improve metanalysis of metabolomics data is exciting. There are several drawbacks/potential complications that are addressed by the authors of the program itself, and outside factors such as progress on standardization of data acquisition and processing.
- I'm struggling to be convinced by lines 245-246. The data we have seen when working on systematic reviews employing metabolomics techniques for specific patient populations is extremely variable- not just with respect to data acquisition and processing but also study design (thinking specifically of inclusion/exclusion criteria and lack of data on important concomitant medications). In reference to lines 244-245, I'd be worried that certain intricacies matter more than others and the overall "essence" of the sample would not necessarily reflect what we think it should. Is there some additional detail you can add to address this, or highlights on curation to know how this could be improved over time?
- You are missing a reference on line 290.
Author Response
Critique: I'm struggling to be convinced by lines 245-246. The data we have seen when working on systematic reviews employing metabolomics techniques for specific patient populations is extremely variable- not just with respect to data acquisition and processing but also study design (thinking specifically of inclusion/exclusion criteria and lack of data on important concomitant medications). In reference to lines 244-245, I'd be worried that certain intricacies matter more than others and the overall "essence" of the sample would not necessarily reflect what we think it should. Is there some additional detail you can add to address this, or highlights on curation to know how this could be improved over time?
Reply by authors: We have added explanations around this issue describing how we plan to capture sample metadata data currently “outside” of the available categories. By doing this, we hope to more concretely determine what categories will, in fact, capture a samples essence. However, we now also highlight the obstacles in ‘perfect’ metadata submissions in the introduction; one of them is the Tragedy of the Commons: biomedical scientists and biologists do not have incentives to spend much time on metadata entries. The benefits of databases are reaped years after data have been uploaded. Basically, SMetaS is an effort of compromises: make it as easy as possible for submitters (e.g. using NLP and automatic curation of freetext), but enable submitters to “add more granularity” for studies such as co-medication. But we do not think we (i.e. SMetaS) can make such mandatory demands: if we would do so, submitters would simply not submit, citing (e.g. for human cohort studies) Hipaa protocols. We may regret this, but in Fiehn's 20+ years of experience as collaborator in studies, such subject metadata are very thorny points of discussion. Surprisingly, things are possible at some NIH institutes (NIA) that are impossible at others (NIDDK,NCI). Those policy decisions cannot be solved by SMetaS.
Critique: You are missing a reference on line 290.
Reply by authors: Fixed
Reviewer 4 Report
The authors have worked on developing a tool (SMetaS (Sample Metadata Standardizer)) that facilitates the standardization of sample descriptions/metadata for downstream programmatic meta-analysis. SMetaS (Sample Metadata Standardizer) combines a classic database with an API and frontend, and enables downstream analysis of submitted studies and samples via string equality for FAIR retrospective use. The tool which might help to integrate all the submitted data (metadata) in the repositories, with future studies without any extra screening and efforts is of utmost need in metabolomics field.
Critics
1. Figure 1, the workflow design should be more detailed.
2. The authors should provide better quality of image for Figure 2. The Figure in its current state is not readable.
3. The authors have discussed a few specific points related to the tool. They should provide more details of the tool. The detailed information in the manuscript will help the metabolomics communities to use and explore the benefits of the tool over other existing tools.
4. The limitations, strengths and future perspectives of the study are not well defined.
Author Response
Critique: At pag 9 Line 290 a reference is missing.
Reply by authors: Fixed
Critique: Figure 1, the workflow design should be more detailed.
Reply by authors: We agree that some readers will want more detail. Therefore, we have provided a highly-detailed workflow as Supplemental Figure 1.
Critique: The authors should provide better quality of image for Figure 2. The Figure in its current state is not readable.
Reply by authors: We have created .tif images during resubmission. These will provide very high-resolution images for the manuscript.
Critique: The authors have discussed a few specific points related to the tool. They should provide more details of the tool. The detailed information in the manuscript will help the metabolomics communities to use and explore the benefits of the tool over other existing tools.
Reply by authors: We have added an additional results section, and largely expanded both the introduction and the discussion section.
The limitations, strengths and future perspectives of the study are not well defined.
Reply by authors: We have expanded the text in the section “Shortcoming and Future of SmetaS”.
Round 2
Reviewer 4 Report
All the comments are well addressed by the authors and overall the manuscript in current form is acceptable for publication in metabolites.